# Glucose-Dependent Insulin Secretion from β Cell Spheroids Is Enhanced by Embedding into Softer Alginate Hydrogels Functionalised with RGD Peptide

**DOI:** 10.3390/bioengineering9120722

**Published:** 2022-11-23

**Authors:** Md Lutful Amin, Kylie Deng, Hien A. Tran, Reena Singh, Jelena Rnjak-Kovacina, Peter Thorn

**Affiliations:** 1School of Medical Sciences, Charles Perkins Centre, University of Sydney, Camperdown, NSW 2006, Australia; 2Graduate School of Biomedical Engineering, The University of New South Wales, Sydney, NSW 2052, Australia

**Keywords:** 3D organoids, spheroids, alginate, insulin secretion, pancreatic beta cells, type 1 diabetes

## Abstract

Type 1 diabetes results from the loss of pancreatic β cells, reduced insulin secretion and dysregulated blood glucose levels. Replacement of these lost β cells with stem cell-derived β cells, and protecting these cells within macro-device implants is a promising approach to restore glucose homeostasis. However, to achieve this goal of restoration of glucose balance requires work to optimise β cell function within implants. We know that native β cell function is enhanced by cell–cell and cell–extracellular matrix interactions within the islets of Langerhans. Reproducing these interactions in 2D, such as culture on matrix proteins, does enhance insulin secretion. However, the impact of matrix proteins on the 3D organoids that would be in implants has not been widely studied. Here, we use native β cells that are dispersed from islets and reaggregated into small spheroids. We show these β cell spheroids have enhanced glucose-dependent insulin secretion when embedded into softer alginate hydrogels conjugated with RGD peptide (a common motif in extracellular matrix proteins). Embedding into alginate–RGD causes activation of integrin responses and repositioning of liprin, a protein that controls insulin secretion. We conclude that insulin secretion from β cell spheroids can be enhanced through manipulation of the surrounding environment.

## 1. Introduction

Type 1 diabetes results from the immunological destruction of pancreatic β cells and the subsequent loss of insulin secretion leads to dysregulated blood glucose levels. The promise of a cell-based solution for the treatment type 1 diabetes is driving exciting research in to producing cells that mimic native β cell glucose sensitivity and insulin secretion. Suspension culture and maturation of stem cells to β-like cells produces organoids ~100–200 µm in diameter [1]. Based on islet transplantation as a yardstick [2], any treatment regime aimed at achieving normoglycemia will require ~1 million of these organoids. However, a major hurdle to a successful treatment is that once implanted these cells would be targeted, and killed, by the patient’s immune system. In the future biological strategies may be developed that prevent immune destruction of these cells but current approaches contain these organoids are either individually, surrounded by a physical barrier [3,4] or packed within macro-devices [5]: both approaches aiming to evade the patient’s immune system [6]. In either case any manipulation of the environment surrounding the organoids aimed at enhancing their function would minimise the number of organoids needed in an implant and optimise therapeutic benefit.

In the normal islet environment β cells interact with other endocrine cells, islet capillary cells and extracellular matrix (ECM) secreted by endothelial cells and pericytes, utilising cell-surface receptors such as cadherins, ephrins, and integrins [7]. Furthermore, additional factors, such as the spatial organisation of β cells to target insulin secretion to the capillaries [8] and cell–cell coordination that define waves of activity through the islet [9], combine to fine-tune glucose-dependent insulin secretion. Importantly, in the intact islet every β cell makes a point of contact with the capillary ECM and also every cell contacts another endocrine cell [7]. This understanding of the spatial organisation and impact of the native environment on β cell function is therefore a useful platform to consider when bioengineering new environments that support advantageous β cell responses.

In this context there has been a long-standing interest in β cell interactions with ECM. These interactions are driven through integrin receptors on the β cells and lead to a plethora of responses, including cell organisation [8], cell survival, replication [10], and importantly enhanced glucose-dependent insulin secretion [11,12]. A major route to study ECM actions uses 2D cell culture [11,12]. This approach has utility but any intervention that affects integrin responses also prevents cell adhesion to the culture dish and therefore destabilises cell–cell interactions, potentially confounding the results. Many previous studies have used β cell lines [13,14] and, given the action of ECM on cell proliferation this may not be relevant to the terminally differentiated stem cell-derived β cells. Furthermore, studies that have used native β cells, commonly use whole islets [15,16,17,18] which are heterogeneous in size (unlike organoids or spheroids) and contain other cell types and endogenous ECM both of which are affected by the addition of ECM to the culture [19].

While these past approaches clearly show the benefits of ECM interactions with β cell function, how this might work in the context of the 3D environment surrounding organoids within an implant device has not been explored. ECM is present in stem cell-derived organoids [20], likely derived from fibroblastic cells that are a by-product of differentiation protocols [1], but this spontaneous production means the ECM is weak and disorganised [20]. Therefore, a directed approach to manipulate the presentation of ECM and control cell organisation in β cell organoids is a promising route to enhance functionality and concomitantly reduce the number of cells required to restore normoglycemia.

Here, we have generated spheroids of defined size and embedded them into alginate hydrogels with, or without conjugated RGD (an ECM peptide motif). We have chosen to use dispersed native β cells in our studies that are then reaggregated to form spheroids. These cells are terminally differentiated and therefore remove complications that might arise from effects of the ECM on stem cell differentiation. The results show that spheroid 3D cell clusters secrete insulin in response to glucose even in alginates without RGD peptides. However, in the presence of RGD there is a significant increase in secretion and reorganisation of β cell structure and an activation of integrin pathways as indicated by phospho-FAK redistribution. To probe this response further we altered the gel stiffness, a factor known to modulate integrin-evoked responses. Our results show that glucose-dependent insulin secretion was still elicited in stiffer gels, but it was significantly reduced compared to the softer gel. We conclude that generating a similar supportive environment around organoids within implant devices is a promising area for development and realising the goal as a treatment for diabetes.

## 2. Materials and Methods

### 2.1. Animal Husbandry

Male C57BL/6 were housed at the Charles Perkins Centre facility in a specific pathogen-free environment, at 22 °C with 12 h light cycles and were humanely killed according to local animal ethics procedures (approved by the University of Sydney Ethics Committee, ethics #2019/1642).

### 2.2. Islet Preparation

Isolated mouse islets were prepared according to a standard method that utilizes collagenase enzymes for digestion and separation from exocrine pancreatic tissue. In brief, a Liberase (TL Research grade, Roche, Basel, Switzerland) solution was prepared in un-supplemented RPMI-1640 (Gibco, Beijing, China) media at a concentration of 0.5 U/mL. Pancreases were distended by injection of 2 mL of ice-cold Liberase solution via the pancreatic duct, dissected and placed into sterile tubes in a 37 °C shaking water bath for 15 min. Isolated islets were separated from the cell debris using a Histopaque (Sigma, St. Louis, MI, USA) density gradient. Isolated islets were maintained (37 °C, 95/5% air/CO_2_) in RPMI-1640 culture medium (Sigma-Aldrich, St. Louis, MI, USA), 10.7 mM glucose, supplemented with 10% FBS (Gibco, Victoria, Australia), and 100 U/mL penicillin/0.1 mg/mL streptomycin (Invitrogen, Victoria, Australia).

### 2.3. Formation of β Cell Spheroids

β cell spheroids were formed using a SpheroFilm (SpheroFilm; HISF-361-3; 300 µm well), InCyto, Cheonan-si, Korea) spheroid-forming PDMS multi-well platform. The SpheroFilm was placed on Petri dishes (60 mm) and prepared according to the manufacturer’s protocol. Briefly, absolute ethanol (7 mL) was added to the Petri dishes, and each well was washed by pipetting. Then, PBS (7 mL) was added, and the wells were similarly washed. The washing step with PBS was repeated twice after which cell culture medium (7 mL) was added, and the Petri dishes were placed in the incubator overnight.

After overnight culture, islets were counted, collected in a microcentrifuge tube, and washed with PBS. Then, TrypLE enzyme (1 µL/islet) was added, and the tube was placed in the water bath for 4.5 min. After this, the islets were dispersed into single cells by pipetting for 30 s, and 1 mL warm medium (RPMI 1640, 15% FBS, 1% pen–strep) was added into it.

The tubes were centrifuged, the medium was discarded, and 1 mL fresh medium was added. The cells were washed one more time, dispersed in the required volume of medium (7 mL for each SpheroFilm), and were immediately placed onto the SpheroFilm after removing the previously added medium from the SpheroFilm-containing Petri dishes. Cells obtained from 160–170 islets were dispersed onto each SpheroFilm, counting approximately 40 β cells in each well, resulting in ~40 µm spheroids. The cells were left on the SpheroFilms for 24 h to aggregate and then were encapsulated in hydrogels.

### 2.4. Preparation of Sodium Alginate Hydrogels

Sterile sodium alginate (Pronova SLG100) and RGD-conjugated sodium alginate (Novatach MVG GRGDSP) were purchased from Novamatrix, Sandvika, Norway. Sodium alginate or RGD-conjugated sodium alginate (0.5 wt%) were weighed in a microcentrifuge tube and dissolved in 300 mM d-mannitol and 1 mM HEPES buffer solution (pH: 7.4) at 2–8 °C overnight. β cell clusters were collected from the dishes by gentle pipetting and centrifuged at 100 RPM for 1 min. The clusters (~350) were added to the sodium alginate solution (100 µL), mixed by gentle pipetting, and added dropwise to the pre-warmed (37 °C) crosslinking solution (50 mM calcium chloride and 1 mM barium chloride in 150 mM d-mannitol and 10 mM MOPS; pH: 7.4) in a 48-well tissue culture plate. The plate was left in the incubator for 10 min, and then the crosslinking solution was replaced by warm medium. The medium was replaced twice after a period of 10 min each. Finally, a small volume of FBS was added (5%; total FBS 15%), and the plate was kept in the incubator (37 °C, 5% CO_2_). Medium was replaced after 24 h.

The stiffness of the alginate-RGD (0.5 wt%) was altered by the addition of 1.5 wt% alginate which was then prepared as above in 300 mM d-mannitol, 1 mM HEPES solution, mixed with the β cell spheroids, and crosslinked.

### 2.5. Mechanical Testing of Alginate Hydrogels

For mechanical testing, alginate hydrogels were prepared in cylindrical moulds (6 mm diameter, 3 mm height) by loading 75 uL of alginate solution in the mould followed by 25 uL of crosslinking solution. After 2 min of incubation at room temperature, all the samples were removed from the moulds and incubated in PBS buffer for 3 h at room temperature before performing the test. All the measurements were performed at a constant crosshead speed (1 mm s^−1^) at 37 °C in the PBS buffer with a 0.1 N pre-load setting using an Instron 5543 mechanical testing machine with a 5 N load cell. Stress–strain curves were obtained, and Young’s modulus was determined from the linear region (10–30% of strain) of the stress–strain curves.

### 2.6. Glucose-Stimulated Insulin Secretion (GSIS) and HTRF Insulin Assay

GSIS was performed after 48 h of encapsulation of the spheroids in hydrogels. GSIS media was Krebs–Ringer bicarbonate solution of pH 7.4 buffered with HEPES (KRBH), plus 2.8 mM glucose (basal) or 16.7 mM glucose (stimulation) composed of: 120 mM NaCl, 4.56 mM KCl, 1.2 mM KH_2_PO_4_, 1.2 mM MgSO_4_, 15 mM NaHCO_3_, 10 mM HEPES, 2.5 mM CaCl_2_ and 0.2% BSA, pH 7.4). This media was used in all the insulin measurements.

Spheroids were washed in warm basal media two times and then placed in fresh basal media for one hour. The basal media was washed out an additional time and then tissues were incubated for 30 min in either fresh basal media or stimulation media. Spheroids were collected at the end of the assay into ice-cold lysis buffer (1% NP-40, 300 mM NaCl, 50 mM Tris-HCl pH 7.4, protease inhibitors) and sonicated. Supernatants and lysates were stored at −30 °C prior to HTRF assay (Mouse ultrasensitive, Cisbio, Huissen, The Netherlands).

### 2.7. Spheroid Fixation and Immunofluorescent Staining

Spheroids were fixed with 4% paraformaldehyde (Sigma-Aldrich) in PBS for 30 min at 20 °C. Samples were stored in PBS at 4 °C prior to immunofluorescent staining. Tissues were incubated in blocking buffer (3% BSA, 3% donkey serum, 0.3% Triton X-100) for a minimum of 40 min at room temperature followed by primary antibody incubation at 4 °C overnight. Sections were washed in PBS (4 changes over 30 min) and secondary antibodies (in block buffer) were added for 4 h at 20 °C. After washing in PBS, tissues were mounted in Prolong Diamond anti-fade reagent (Invitrogen). Primary antibodies used for this study were: anti-insulin (Dako Cytomation, A0564), anti-phosphorylated FAK (Cell Signalling Tech 8556S), anti-liprin alpha1 (Proteintech 14175-1-AP). All primary antibodies were diluted 1/200. Secondary antibodies were highly cross absorbed donkey or goat antibodies (Invitrogen) labelled with Alexa 488, Alexa 546, Alexa 594, or Alexa 647. All were used at a 1/200 dilution. DAPI (Sigma, 100 ng/mL final concentration) was added during the secondary antibody incubation. Confocal imaging was performed on a Leica SP8 microscope with a 63X oil immersion objective. The fluorescence intensity in regions of interest along the cell along the cell membrane was used to quantify the images.

### 2.8. Statistical Analyses

All numerical data are presented as mean +/− standard error of the mean. Statistical analysis was performed using Microsoft Excel and GraphPad Prism. Data sets with two groups were subjected Student’s t-test, unpaired, equal variance. Significance is indicated as follows: * *p* < 0.05, ** *p* < 0.01, *** *p* < 0.001, **** *p* < 0.0001.

## 3. Results

### 3.1. Production of β Cell Spheroids of Consistent Diameter

In all experiments we used isolated mouse islets which were cultured intact overnight (20–22 h) and then dispersed to single cells (Figure 1A). This resulted in a preparation rich in endocrine cells and effectively devoid of endogenous ECM and endothelial cells. These isolated cells were then resuspended and aliquoted across a Spherofilm multi-well PDMS platform, and after a day (24 h) in culture led to the generation of spheroid clusters of endocrine cells of defined size (Figure 1A). We used clusters of 40 cells with the aim of producing small spheroids where most cells had an outward face.

At day 2 after isolation from the mouse the small spheroids (Figure 1B) were harvested and resuspended in alginate which was then cross linked by dropping into a divalent-rich solution into alginate beads of ~700 µm in diameter (Figure 1C). We used either alginate (0.5%) alone or alginate conjugated with RGD. The diameter of the spheroids embedded within the alginate was measured in either alginate alone or with RGD and were around 40 µm in diameter with no significant difference in spheroid diameter between the two different gels (Figure 1D).

### 3.2. Glucose-Dependent Insulin Secretion from Spheroids

The spheroids embedded in alginate beads were cultured for 2 days (42–44 h) after which their function was tested with a glucose challenge. The protocol used a low glucose (2.8 mM) pre-incubation period for 1 h, a wash, and then a step increase to 16.7 mM glucose. In both alginate and alginate-RGD an increase from 2.8 mM to 16.7 mM glucose for 30 min led to a significant increase in insulin secretion (Figure 2A) and analysis of the secretory index (fold increase in insulin secretion) showed that this was significantly greater in the presence of the RGD motif (Figure 2B).

### 3.3. Effect of Alginates on β Cell Structure within Spheroids

To test whether alginate-RGD acted through the activation of integrins on the β cell surface we performed immunostaining on the embedded spheroids with an integrin β1 antibody (Figure 3). In both alginate alone and alginate-RGD, integrin β1 was present throughout the cytosol as shown in the representative image (Figure 3A,D), in a line scan analysis of image intensity (Figure 3B,E) and in an analysis of the inner vs. outer area of the cluster (Figure 3C,F). We noted here that immunolocalization of integrin β1 shows this protein was present throughout the cells but did not indicate whether focal adhesions had been activated. To this end, recent work shows that integrin/focal adhesion activation has a direct effect on the stimulus secretion cascade and acts on synaptic scaffold proteins and the glucose-dependent calcium response to enhance insulin secretion [8]. To determine if this was the case in these spheroids we immunostained for the synaptic scaffold protein liprin. The results showed that spheroids embedded in alginate alone showed patchy staining throughout the cell cluster (Figure 3A–C) but a significant enrichment in the outer membrane when embedded in alginate-RGD (Figure 3D–F).

To further probe the cellular response to embedding into alginates we used a phosphorylation-specific antibody to focal adhesion kinase (phospho-FAK) and compared the fluorescence at the membrane along the cell periphery (which contacts the alginate) with the fluorescence at the internal membrane (which contacts other endocrine cells). In spheroids embedded within alginate alone we observed no distinct distribution but in those embedded in alginate-RGD we saw a significant enrichment of phospho-FAK staining on the outer surface demonstrating integrin/focal adhesion activation (Figure 4).

Another measure of cell orientation is the organisation of cell-to-cell junctions, typified by cadherins. Immunostaining showed that E-cadherin was enriched where the β cells within the spheroids touched each other (Figure 5) and was not found where the cells contacted the surrounding alginate-RGD.

We conclude that the β cells within the spheroids respond specifically to the alginate-RGD as measured by an enhanced glucose-dependent insulin secretion and structural changes (summarised in Figure 5B) to the outer cells that contact the alginate-RGD.

### 3.4. Effect of Alginate Stiffness on Glucose-Dependent Insulin Secretion

If the alginate-RGD was acting through integrins we would expect it to be influenced by alginate stiffness since this is a parameter that is well known to modulate integrin responses [21,22] and has been shown to enhance insulin expression in a β cell line [14]. To test this, we created alginates where the alginate-RGD concentration was kept the same, but the proportion of alginate was increased from 0.5% to 1.5%. Compression testing and measurements showed differences in the mechanical properties between the two gels and a significantly larger compression modulus in the 1.5% alginate compared to the 0.5% (Figure 6). This confirmed the difference in the mechanical stiffness of the gels which were then used to embed the spheroids as shown in Figure 1.

After 2 days in culture the embedded spheroids were tested with a glucose challenge. An increase in glucose from 2.8 mM to 16.7 mM, for 30 min elicited a significant increase in insulin secretion in both the soft-RGD gel (same data as in Figure 2) and the stiff-RGD gel (Figure 7A). However, the increase in secretion was much lower in the stiff-RGD gel and showed a significant decline in the stimulation index (Figure 7B). We conclude that gel stiffness is a critical co-factor that modulates the RGD-evoked responses in β cells and controls insulin secretion.

## 4. Discussion

Although there have been numerous studies showing the benefit of ECM interactions on β cell function, how to best translate these to the design of an internal environment of macro-implant devices has not been clear. Our work used small spheroids consisting of terminally differentiated native β cells and tested the impact of embedding these spheroids into functionalised alginate hydrogels.

We showed that spheroids embedded in alginate alone, robustly secreted insulin in a glucose-dependent manner indicating that the essential stimulus-secretion pathway was intact. This was consistent with previous work that shows even single β cells retain the capacity to secrete [23]. However, in our experiments embedding the spheroids into alginate-RGD led to a significant and approximately two-fold increase in glucose-dependent insulin secretion. Through immunostaining we showed that alginate-RGD triggered a reorganisation of the β cells with an orientation of phospho-FAK, and the presynaptic scaffold protein liprin, to give an enrichment at the outer edge of the outer cells in the spheroids. Consistent with an integrin-mediated effect we showed that altering the stiffness of the alginate-RGD had a significant impact on glucose-dependent insulin secretion, with softer gels leading to enhanced secretion. Together this data indicates that the alginate-RGD triggered an integrin response in the β cells and an organisation of the secretory machinery (i.e., orientation of liprin). This structural response to alginate-RGD was paralleled by an increase in secretion indicating an up-regulation of the stimulus-secretion pathway. Exactly how this increase occurs is not shown by our data but we can speculate that the structural reorganisation impacts on the final stages of insulin granule fusion, potentially by the positioning sites of granule fusion close to the sites of calcium entry—something we have recently shown to occur in the native islet [8]. If this was the case, we would expect to see granule fusion and calcium entry along the outer edges where the cells contact the alginate-RGD. That is beyond the scope of this paper but would be an interesting way of testing these predictions.

Two forms of contact predominate β cell responses in the native islet. There are cell–cell cadherin-based contacts with adjacent endocrine cells [24] and cell to extracellular matrix integrin-based contacts with the matrix secreted by endothelial cells and pericytes [7,25]. Within the islet most β cells contact the capillary matrix and this defines a basal domain with the cells, with lateral domains forming through cadherin contacts between adjacent endocrine cells [26]. In this way β cells have a polarised structure [26] that appears to underlie functional polarisation, such as targeting insulin granule fusion to the basal domain [8,27]. Previous work has co-cultured endocrine cells with endothelial cells [28] to attempt to recapitulate the native organisation of the islet. However, to our knowledge, although some benefit has been observed, it has not been possible to reproduce the spatial arrangement of these cell types and the polarisation of the β cells. The spheroid structure we describe here means that β cells make cell–cell cadherin contact and, at least in small spheroids, most cells will contact the surrounding RGD-alginate and therefore make integrin contact. These mimic the two contacts made by β cells in native islets and therefore forms a minimum approach to designing an artificial environment that establishes β cell polarity and, as we now show, enhanced insulin secretion. Overall, we conclude that β cell organoid function is significantly affected by embedding in biofunctionalized alginates and that this is a promising route to enhance insulin secretion from cell clusters within an implant.

## Figures and Tables

**Figure 1 bioengineering-09-00722-f001:**
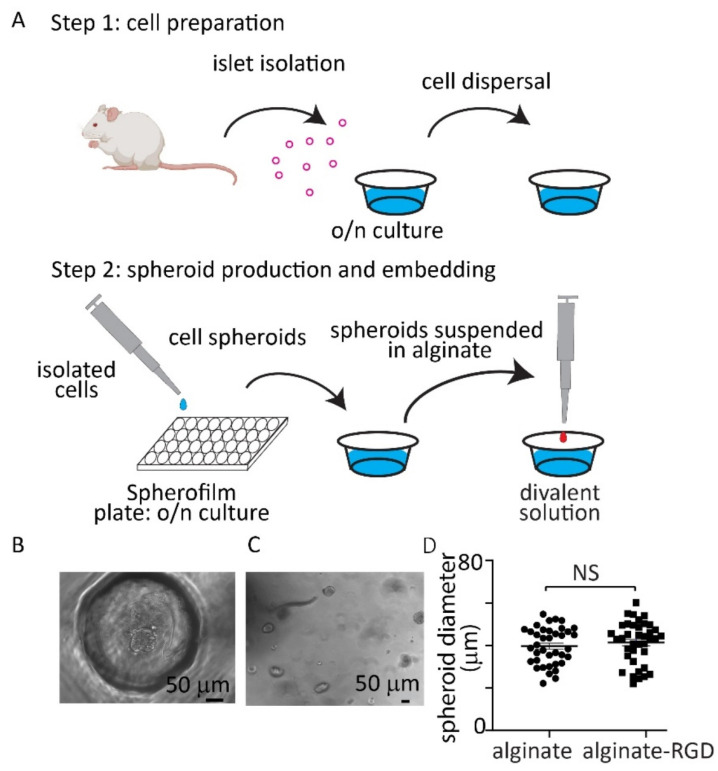
Preparation of spheroids of uniform size from native mouse β cells. (**A**) Cartoon representation of protocol where pancreatic islets were harvested from mice, cultured overnight and then dispersed to single cells. These cells were then aggregated in low-adhesion wells to form spheroids of controllable size that were then embedded into alginate beads. (**B**) Example image of the formation of spheroids within the PDMS wells. (**C**) Example image of spheroids within alginate bead. (**D**) Analysis of spheroid diameter within alginate (0.5%) alone or alginate (0.5%) + RGD beads showed no significant differences (n = 38 clusters in both alginates, Student *t* test *p* = 0.43, cells from n = 3 mice).

**Figure 2 bioengineering-09-00722-f002:**
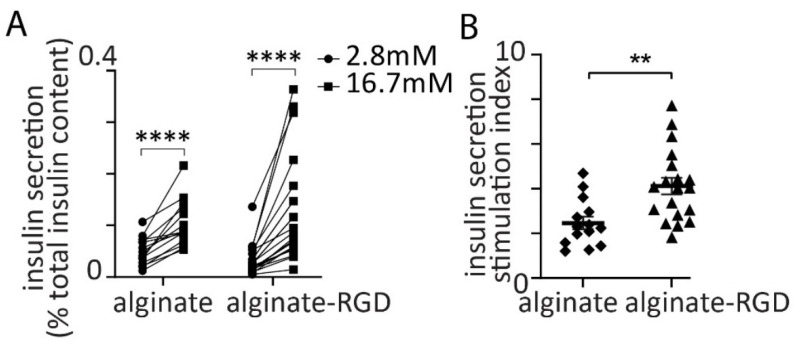
Embedding spheroids in alginate-RGD enhanced glucose-induced insulin secretion and polarised individual β cells. (**A**) A step increase in glucose from 2.8 mM to 16.7 mM induced a significant increase in insulin secretion from spheroids in alginate (n = 14 beads, Student *t* test *p* < 0.001, **** *p* < 0.001) and in alginate-RGD (n = 23 beads, Student *t* test *p* < 0.001). (**B**) A comparison of the insulin stimulation index (fold increase) in insulin secretion induced by high glucose showed a significant enhancement with the presence of RGD (Student *t* test *p* < 0.01, ** *p* < 0.01 ).

**Figure 3 bioengineering-09-00722-f003:**
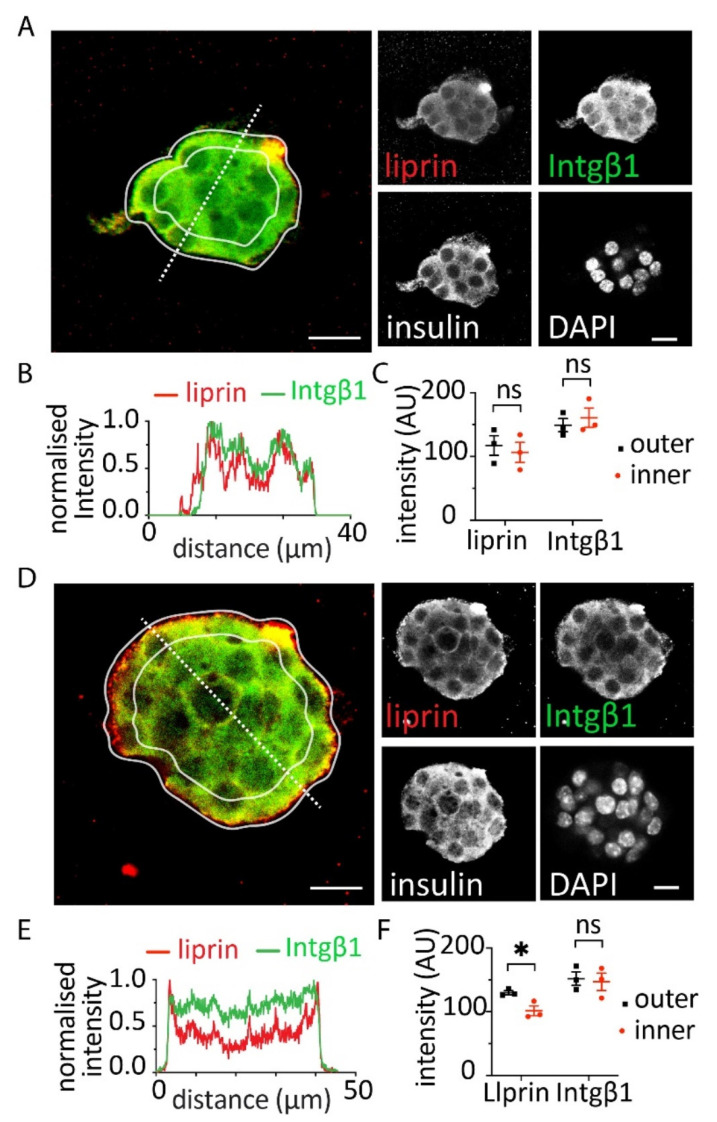
Immunostaining showed that cells within the spheroids reorganised in response to the RGD domain. Immunostaining of spheroids embedded in (**A**–**C**) alginate showed evidence for integrin β1 and liprin throughout the cell cytosol. This was observed in the images (**A**), line scans of fluorescence intensity drawn through the spheroids (**B**) and comparison of fluorescence intensity in the inner vs. outer areas of the cluster (**C**). (**D**–**F**) In contrast, spheroids embedded in alginate + RGD showed evidence of enrichment in liprin on the outer surface of the outer cells, although no enrichment of integrin β1 was observed. This was observed in the images (**D**), line scans of fluorescence intensity drawn through the spheroids (**E**) and comparison of fluorescence intensity in the inner vs. outer areas of the cluster (**F**). Counterstaining with insulin identified the cells as β cells. Scale bars 5 µm (* Student *t* test *p* < 0.05).

**Figure 4 bioengineering-09-00722-f004:**
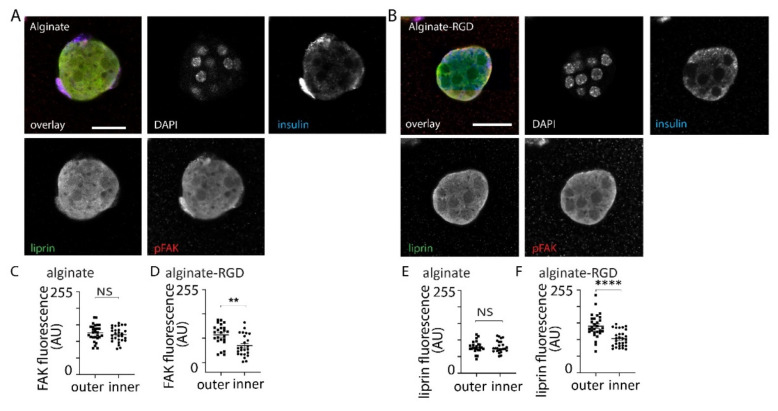
Immunostaining showed activation of focal adhesion kinase, measured with a phospho-specific antibody, in response to the RGD domain. (**A**,**B**) Example images from spheroids within alginate alone or alginate-RGD showed the distribution of liprin and phospho-FAK. (**C**,**D**) Image analysis comparing the fluorescence intensity of pFAK staining at the outer membranes with staining on the inner membrane demonstrating no difference in spheroids embedded in alginate (n = 28 cells, Student *t* test *p* = 0.41) but a significant difference in alginate-RGD (n = 24 cells, Student *t* test *p* < 0.01, ** *p* < 0.01 ). (**E**,**F**) Image analysis comparing the fluorescence intensity of liprin staining at the outer membranes with staining on the inner membrane demonstrating no difference in spheroids embedded in alginate (n = 24 cells, Student *t* test *p* = 0.92) but a significant difference in alginate-RGD (n = 29 cells, Student *t* test *p* < 0.001, **** *p <* 0.001). Scale bars 20 µm.

**Figure 5 bioengineering-09-00722-f005:**
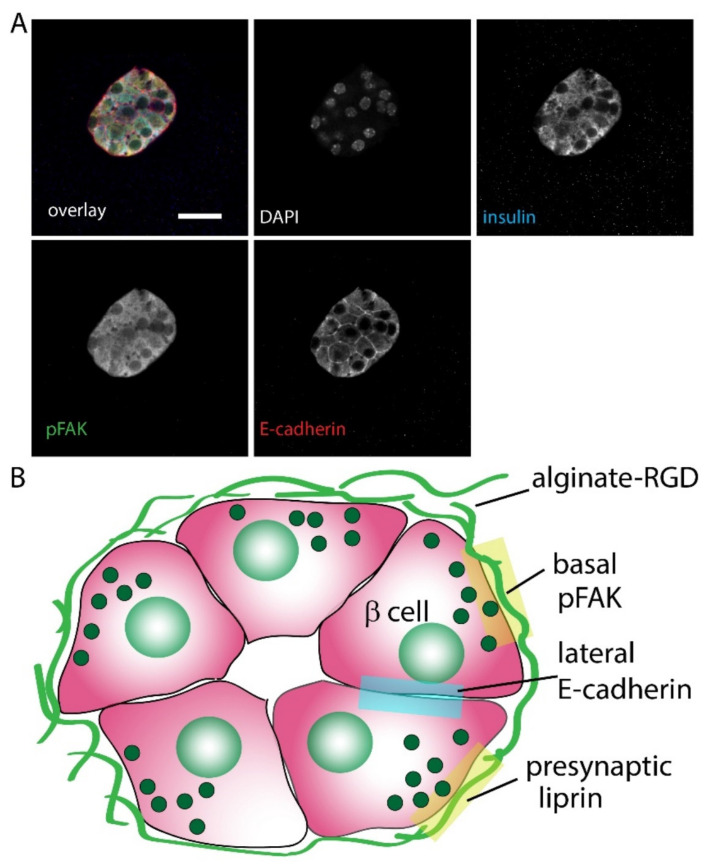
Immunostaining showed formation of E-cadherin junctions between the β cells. (**A**) Example images from clusters within alginate-RGD showed that E-cadherin was enriched where individual β cells contacted each other and was excluded from the outer domain where the cells contacted the gel. (**B**) Cartoon representation of the β cell spheroids showing the organisation of cell structure and orientation to the RGD alginate.

**Figure 6 bioengineering-09-00722-f006:**
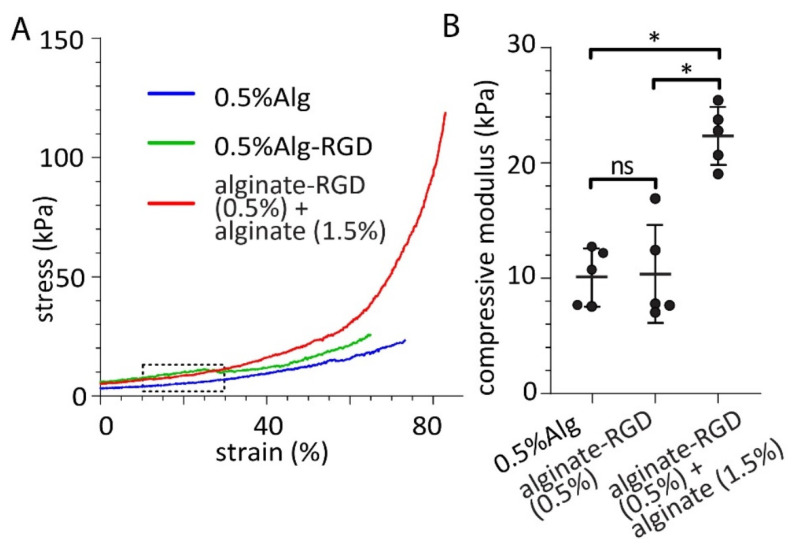
Mechanical properties of alginate hydrogels. (**A**) Stress–strain curves of 0.5% alginates without RGD and with RGD, and 0.5% alginate with RGD plus 1.5% alginate. (**B**) The compressive modulus of the different alginate hydrogels measured from the linear region (10–30% strain—dashed line in (**A**)). Statistical analysis, one-way ANOVA with Turkey post hoc test. * *p* < 0.05, ns = non-significant.

**Figure 7 bioengineering-09-00722-f007:**
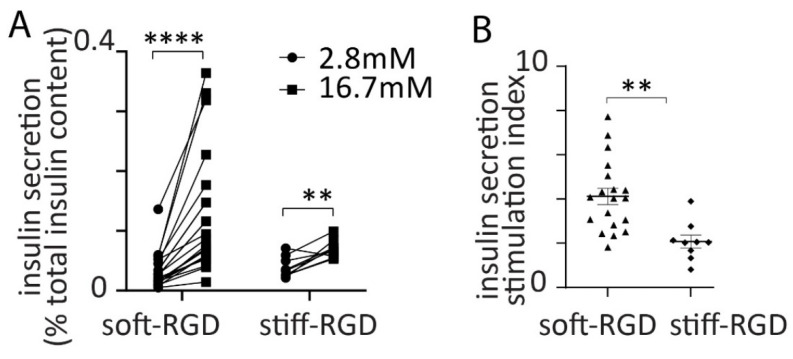
Effect of alginate stiffness on glucose-induced insulin secretion. (**A**) Spheroids were embedded in soft-RGD alginate or stiff-RGD alginate and then subjected to a stepped increase in glucose from 2.8 mM to 16.7 mM. This induced a significant increase in insulin secretion from spheroids both in the soft-RGD (n = 23 beads, Student *t* test *p* < 0.0001 -same data as in Figure 2A, **** *p* < 0.001) and in the stiff-RGD (n = 9 beads, Student *t* test *p* < 0.001). (**B**) A comparison of the insulin stimulation index (fold increase) in insulin secretion induced by high glucose showed an enhancement in softer alginate-RGD (Student *t* test *p* < 0.01, ** *p* < 0.01).

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
