# Peer review of "Glucose-Dependent Insulin Secretion from β Cell Spheroids Is Enhanced by Embedding into Softer Alginate Hydrogels Functionalised with RGD Peptide"

_bioengineering, 2022, doi:10.3390/bioengineering9120722_

Round 1

Reviewer 1 Report

Review to bioengineering-2033815

It is highly recommended that the authors should not use organoids (especially βcell organoids) to rub the hot spot. I think it should be regarded as unresponsible or unwise for the authors and journal used organoids in this manuscript. This cell aggregates and organoids are two different things, their definition is completely different. It will cause confusion and mislead for many researchers. I can accept it as a good in vitro model based on terminally differentiated β cell aggregates.

They compared the enhanced glucose-dependent insulin secretion by embedding β cell aggregates in softer alginate hydrogels functionalised with RGD peptide. Normally, RGD is often used to enhance biocompatibility of materials by grafting and modifying. 

Author Response

We thank the referee for their comments and have taken the chance to make revisions to the manuscript.

The reviewer questions our use of the term "organoids". Unfortunately, there is no definition of the word. We presume the referee's feels organoids implies a 3D structure produced from stem cells but, the term is also used for structures produced from cell lines and adult cells.

Notwithstanding these arguments, in response to the referee, we have changed the description of our 3D structures from "organoids" to "spheroids".

Reviewer 2 Report

The paper by Amin and colleagues presents a well conducted structural and functional characterization of beta cells-based organoids. Whereas data are clearly shown, more explanations on the implications of this work should be provided in the discussion.

Additional suggestions to improve the manuscript are given below:

-line 104: include the authorization number

-line 193 and result section: data must be expressed as mean and standard deviation, not standard error

Minor:

-line 49: the indication of 1M is confusing, since M is usually employed for Molar

-check for some typos (i.e lines 46,96)

Author Response

We thank the referee for their comments and have made revisions throughout the manuscript. We have addressed all the minor changes and typos identified by the referee.

The referee suggests we use standard deviation rather than standard error. Both of these are descriptors of the data, one referring to the sample error and the other an estimate of the error in the population. Neither is a better descriptor and therefore we ask to leave this as standard error. We make the point that the critical information for the reader are the statistical tests we have performed. Furthermore, in our graphs we use the preferred approach of scatter plots that show each data point rather than histograms.

Round 2

Reviewer 1 Report

The spheroids are better than organoids.  It should be a good in vitro model based on terminally differentiated β cell spheroids with enhanced glucose-dependent insulin secretion. The authors have fully considered my suggestions. I advise to accept it in present form.